# Rates of Suicide Ideation and Associated Risk Factors Among Female Secondary School Students in Iraq

**DOI:** 10.3390/healthcare13111260

**Published:** 2025-05-27

**Authors:** Saad Sabet Alatrany, Molly McCarthy, Ashraf Muwafaq Flaiyah, Emma Ashworth, Hasan ALi Sayyid ALdrraji, Abbas Saad Alatrany, Dhiya Al-Jumeily, Sarmad Nadeem, Jo Robinson, Pooja Saini

**Affiliations:** 1Department of Clinical Psychology, Imam Ja’afar Al-Sadiq University, Baghdad 10001, Iraq; saad.alatrany@ijsu.edu.iq (S.S.A.); ashraf.flaiyah@ijsu.edu.iq (A.M.F.); 2Ibn Reshed College of Education for Human Sciences, University of Baghdad, Baghdad 10071, Iraq; hassan.sayid@ircoedu.uobaghdad.edu.iq; 3School of Psychology, Liverpool John Moores University, Liverpool L3 3AF, UK; m.mccarthy1@ljmu.ac.uk (M.M.); e.l.ashworth@ljmu.ac.uk (E.A.); 4Baghdad Centre for Psychosocial Support, Baghdad 10091, Iraq; 5Biomedical Informatics College, University of Information Technology and Communications, Baghdad 10066, Iraq; assa9@leicester.ac.uk; 6Department of Cardiovascular Sciences, University of Leicester and the NIHR Leicester Biomedical Research Centre, Glenfield Hospital, Leicester LE3 9QP, UK; 7School of Computer Science and Mathematics, Liverpool John Moores University, Liverpool L3 3AF, UK; d.aljumeily@ljmu.ac.uk; 8Pennine Care NHS Foundation Trust, Lancashire OL6 7SR, UK; sarmad.nadeem@nhs.net; 9Centre for Youth Mental Health, The University of Melbourne, Parkville, VIC 3010, Australia; jo.robinson@orygen.org.au

**Keywords:** school, female, suicide, self-harm, suicide prevention

## Abstract

Background: The suicide rate among Iraqis is rising, with many analysts attributing it to political instability, exposure to trauma, economic hopelessness, social stigma surrounding mental health as well as cultural and societal pressures. However, the prevalence of suicidal ideation and associated risk factors in Iraqi youth is unknown, requiring urgent attention and effective public health initiatives. Thus, the aim of this study was to explore rates of suicidal ideation and associated risk factors among female secondary school students in Baghdad, Iraq. Method: A cross-sectional study was conducted, utilising quantitative survey data collected in four girls’ secondary schools across Baghdad, Iraq, between August and December 2023. The survey consisted of questions relating to their demographic characteristics (age, gender, school) and a series of measures pertaining to participants’ levels of suicidal ideation, as well as factors commonly identified in the literature as predictors of suicide. Results: Four-hundred and two female participants took part. Participants were aged between 13 and 17 years (*M* = 15.50; *SD* = 1.22). In total 11.3% of the students scored in the at-risk range for suicidal behaviour and only 20.1% (*n* = 91) said they had not had some thoughts of suicide in the previous two weeks. Previous diagnoses of anxiety, high levels of depression and hopelessness, and poor quality of life were significant risk factors for suicidal ideation. On average, students reported moderate levels of depression and high levels of hopelessness. Conclusions: Female Iraqi secondary school students experience high levels of suicidality, alongside several other known risk factors for suicide ideation. However, a limitation of this study is that cross-sectional designs limit causal interpretation. Findings emphasise the importance of developing targeted school-based interventions to support students’ mental health. Increasing research and attention in this area is vital to not only improving the mental health of students in Iraq but also reducing the stigma around mental health and suicide. Future policies should include specific mental health support for those young people affected by conflict, displacement and family loss, integrating trauma-informed care into both mental health and educational services.

## 1. Background

Suicide is the second leading cause of death amongst youth aged 10 to 34 worldwide. Approximately 703,000 people die by suicide annually, and 20 suicide attempts occur for every suicide [1]. Therefore, practical strategies to prevent or reduce suicide are necessary. Data from the Iraq National Study of Suicide revealed that the crude suicide rate was 10.9/100,000 in 2016 [2] compared to 7.3/100,000 in 2017 in the UK [3], with statistics suggesting that deaths by suicide have been increasing each year since [4]. Among those who died, 36.6% were below the age of 20 [5]. There are particular concerns of a rising suicide rate among Iraqi youths, with many analysts suggesting it may be due to the economic hopelessness so many have experienced during childhood. However, little is known about the current suicide rate in Iraqi youth, or the key risk factors. The prevalence of suicidal ideation and associated risk factors in Iraq, therefore, requires urgent attention and effective public health initiatives, particularly targeting youth, who seem to have been omitted to date.

Basra recorded the highest rates of suicide or suicide attempts across Iraq, with the ages of those dying by suicide ranging from 18 to 30 years old [6]. However, the rates are generally considered to be an underestimate, as many people keep deaths by suicide hidden for social reasons. For instance, doctors in Iraq have reported facing pressure from relatives and tribal authorities to prevent post mortem examinations from investigating the cause of death for two key reasons: (1) due to socio-religious beliefs that the autopsy would damage the body, and (2) to cover up the real reason behind the death, as the family refuse to disclose it due to social stigma and religious concerns [7].

In Iraq, several other factors are thought to contribute to levels of suicide ideation, including intimate partner problems, physical health conditions, financial challenges, and legal issues [7]. Others are personal or family experiences of violence, including child abuse, neglect, and domestic violence, as well as family history of suicide and broader community conditions, such as high crime rates and violence [8]. The COVID-19 pandemic is also known to have contributed to an increase in several risk factors, such as rates of domestic violence in the country, with one survey reporting that currently one in five Iraqi women experience domestic violence [9].

Furthermore, the political situation in Iraq over the last two decades may present with several unique risk factors for adolescents and young adults, with particular reference to the Iraq mental health crisis [10]. For instance, many Iraqi children and their families have suffered with mental health scars caused by past conflicts, wars, and economic hostilities, including control by the Islamic State (ISIS) [11]. As a result, a study conducted in Mosul across eight secondary schools reported 30% had symptoms of post-traumatic stress disorder (PTSD), with rates higher in older adolescents [12]. Amplifying the existing mental health concerns among Iraqi youth is the lack of mental health support and treatment across the country [13]. These have been further compounded by the stay-at-home restrictions and limited movements required to curb the spread of COVID-19. Concerns of increased suicide ideation and associated risk factors have been highlighted by public health networks globally [9].

Iraqi females are thought to be disproportionately affected by suicidal ideation, self-harm and suicide attempts, often noted to be driven by factors such as familial conflict, economic dependence, and the limitations placed on women’s roles within society [14]. For example, in a recent systematic review [7] young females represented higher levels of suicide attempts and ideation using potentially lethal methods such as self-burning, thought to be due to high rates of depression, as well as community and domestic violence. Furthermore, a comprehensive survey conducted in Iraq by the WHO in collaboration with the Iraqi Ministry of Health [15], the Iraqi Mental Health Survey (IMHS) 2006–2007, revealed that 5.4% of females had seriously considered suicide compared to 2.5% of males. Females were also nearly three times more likely to engage in suicide attempts. These findings are consistent with broader global trends where females, particularly adolescents and young women, are more vulnerable to non-lethal self-harm and suicide attempts, often to cope with psychological pain, depression, and trauma [4]. As such, work centred around female adolescents is imperative to ensure appropriate targeted support is on offer.

The current study aimed to explore rates of suicidal ideation and associated risk factors among female secondary school students in Baghdad, Iraq, a previously overlooked public health concern.

## 2. Methods

### 2.1. Design

The current cross-sectional study utilised quantitative survey data collected in secondary schools across Baghdad, Iraq, between August and December 2023.

### 2.2. Participants and Recruitment

Four female secondary schools were recruited to participate. Baghdad is divided into two parts, known as Karkh and Rusafa, by the Tigris River. Consequently, two female schools were selected from each part to ensure representation of schools across the city. Data collectors visited the four selected schools and, in coordination with the school principals, selected a maximum of two classrooms per school (with an average of 50 students per classroom). Students were invited to complete paper-based surveys in school, which included all measures stated below. If the parents/carers consented to their child taking part, students were invited to complete paper-based surveys in school. To maintain participant privacy, female data collectors were assigned to female schools. Informed assent was sought from the students, who were asked to tick a box at the beginning of the questionnaire if they consented to taking part. Students were informed that participation was voluntary, and they could leave before or during the completion of the questionnaires without any negative consequences. Some students chose to leave before starting or did not complete the questionnaires (n = 27; 6%). The study received ethical approval on 15 August 2023 Imam Ja’afar Al-Sadiq University, College of Arts Ethical Approval Committee (Ref: 2023-01).

### 2.3. Measures

The survey consisted of two parts: Part 1 asked participants questions relating to their demographic characteristics (age, gender, school), while Part 2 presented a series of measures pertaining to participants’ levels of suicidal ideation, as well as factors commonly identified in the literature as predictors of suicide (outlined below). All measurements have been validated for use with adolescent populations except for the brief two-item hopelessness measure.

#### 2.3.1. Suicidal Ideation

Suicidal ideation was assessed via the Suicidal Ideation Attributes Scale (SIDAS) [16]. The SIDAS is a self-report measure designed to screen individuals in the community for the presence of suicidal thoughts and assess the severity of these thoughts, over the past month. It comprises five items, each targeting an attribute of suicidal thoughts: frequency, controllability, closeness to attempt, level of distress associated with the thoughts, and impact on daily functioning. Responses are measured on a 10-point scale. Total SIDAS scores are calculated as the sum of the five items, with controllability reverse scored, and with total scale scores ranging from 0 to 50. A higher total score reflects more severe suicidal thoughts. Scores over 21 represent the clinically significant threshold for high-risk suicide ideation.

#### 2.3.2. Depression and Suicidal Thoughts and Plans

Depression symptoms were assessed using the Patient Health Questionnaire-9-item version (PHQ-9) [17]. Participants are asked to indicate how often they have been bothered by nine problems over the past two weeks. Each item is rated on a four-point Likert scale ranging from 0 (“not at all”) to 3 (“nearly every day”). Scores are summed such that the potential range is 0–27, with higher scores indicative of greater distress. Clinical thresholds are provided, indicating ‘mild’, ‘moderate’, ‘moderately severe’ and ‘severe’ depressive symptoms. The final question of the PHQ-9 asks participants to indicate the number of days they have had thoughts that they would be better off dead or hurting themselves in some way. If they indicated any suicidal thoughts, participants were presented with a purpose-designed item, asking them to indicate which of the following describes the level of suicidal ideation they were experiencing: “Mild suicidal thoughts with no plan or intent to act”; “Moderate suicidal thoughts with a rough plan and some intent”; “Severe suicidal thoughts with a specific plan and intent to act”.

#### 2.3.3. Hopelessness

Hopelessness was assessed using the Brief-H-Pos, a two-item positively worded measure of hopelessness [18]. Respondents indicate agreement on a five-point scale (range 2–10), with lower scores indicating higher hopelessness.

#### 2.3.4. Help-Seeking Intentions

Intentions to seek help were assessed using part two of the General Help Seeking Questionnaire (GHSQ) [19]. The GHSQ presents participants with a list of potential sources of help and asks them to indicate the likelihood that they would approach that source if they were experiencing suicidal thoughts on a five-point scale (very unlikely-very likely). Higher scores indicate greater levels of intended help-seeking.

#### 2.3.5. Health-Related Quality of Life

Health-related quality of life was assessed using the Child Health Utility–9 (CHU9D) [20]. The CHU9D is a multi-attribute utility instrument suitable for young people aged 7–17 years. The questionnaire has nine items, each with a five-level response category that assesses a young person’s functioning “today” across various domains. The total score ranges from 0 to 40, with higher values indicating greater behavioural and emotional difficulties.

#### 2.3.6. Suicide Literacy

Suicide literacy was assessed using an adapted version of the Literacy of Suicide Scale (LOSS) short form to ascertain participants knowledge about suicide [21]. This contains nine statements about suicide and asks participants to rate whether they believe it is true, false, or they do not know. The scale provides a total literacy score (percent correct), where higher scores indicate greater suicide literacy.

### 2.4. Procedure and Analytic Strategy

A pre-testing pilot was conducted to test the translation of the survey from English to Arabic on a small group similar to the target population (n = 7). The survey was then distributed within schools. See Box 1 for more information on the translation process.

Prior to analysis, participants’ responses on the paper-based surveys were scanned using Remark Office OMR software Version 11 by data entry personnel. The file was then imported into IBM SPSS Version 29 for analysis.

This was an exploratory analysis, whereby descriptive statistics were conducted to identify the socio-demographics of the sample and the factors characteristic of students experiencing suicide ideation. To achieve this, we conducted the following set of analyses. Descriptive analysis (e.g., means, standard deviations, frequencies, percentages) were used to characterise the overall sample in terms of age, gender, ethnicity, academic status, and other socio-demographic variables. Next, chi-squared analyses were used to assess associations between categorical variables (e.g., gender and suicidal thoughts), as well as independent sample *t*-tests and one-way ANOVAs to compare means of continuous variables (e.g., levels of stress, social support) across groups defined by the presence or absence of suicide ideation or across multiple categorical groups (e.g., academic year). Association analyses using Spearman’s Rank correlation were also employed to examine relationships between continuous and ordinal variables (e.g., perceived stress and suicidal ideation scores). Regression analyses were then conducted with suicide ideation and suicidal thoughts as outcome variables to identify significant predictors. We did not conduct Bonferroni corrections for multiple comparisons based on recommendations from Armstrong [22] and Rothman [23] that corrections for multiple comparisons in exploratory studies are not required, due to the increased likelihood of Type 2 errors.

Box 1Translation process.**1.** 
**Formation of Translation Teams:**
A translation team was established, comprising a professional English translator and a professor specialising in clinical psychology who is proficient in English.
**2.** 
**Translation of Questionnaires:**
The team translated the questionnaires from English to Arabic.
**3.** 
**Language Review:**
The translated questionnaires were reviewed by an Arabic language specialist to ensure linguistic accuracy and appropriateness for secondary school students.
**4.** 
**Back-Translation:**
A second team was formed, consisting of a professional English translator and a professor specialising in clinical psychology who is proficient in English. This team performed a back-translation of the Arabic questionnaires into English.
**5.** 
**Review and Finalisation:**
A meeting was held between the two translation teams to discuss feedback regarding the translations and to agree on a finalised Arabic version of the questionnaires.


## 3. Results

### 3.1. Demographic Characteristics

A total of 452 female participants took part. Participants were aged between 13 and 17 years (*M* = 15.50; *SD* = 1.22). The most common mental health diagnosis reported by students was anxiety (n = 190; 42%), followed by depression (n = 99; 21.9%) and obsessive-compulsive disorder (OCD; n = 77; 17.0%). The prevalence of neurodivergent conditions was broadly in-line with international rates, with 2.0% (n = 9) reporting an autism diagnosis and 11.1% (n = 50) reporting a diagnosis of attention deficit hyperactivity disorder (ADHD). See Appendix A Table A1 which provides the breakdown of mental health diagnosis.

### 3.2. Suicidal Ideation (SIDAS)

Mean scores for suicide ideation on the SIDAS were 8.45 (SD = 10.30) out of a maximum score of 50 (range = 0–50). Seven participants (1.5%) had the maximum possible score. Responses to the individual items are presented on Appendix A Table A2.

A total of 11.3% of the students (n = 51) scored in the at-risk range for suicidal behaviour (a score ≥ 21). This was consistent across age groups. A logistic regression was performed to identify significant predictors of high-risk SIDAS scores. The overall model was statistically significant (c^2^ (1) = 186.127, *p* < 0.001), explaining 17.5% of the variation (Nagelkerke R^2^). Depression (*p =* 0.001), quality of life (*p* = 0.003), and hopelessness (*p* = 0.044) were significant predictors of high-risk SIDAS scores. Table 1 provides further details.

A logistic regression was also performed to ascertain the effects of previous mental health diagnosis on at-risk SIDAS scores. The overall logistic regression model was statistically significant (c^2^ (1) = 186.127, *p* < 0.001), explaining 10.4% of the variation (Nagelkerke R^2^). Those with a mental health diagnosis of anxiety were 1.9 times more likely to score ≥ 21 on SIDAS (*p* = 0.052). Similarly, students with any diagnosis of a personality disorder were 3.6 times more likely to score ≥ 21 (*p* = 0.027). No other predictors were significant.

### 3.3. Suicidal Thoughts and Behaviours

The final question on the PHQ-9 relates to suicidal thoughts—‘thoughts that you would be better off dead’. Nearly half of the sample (n = 220; 48.7%) disclosed they had these thoughts ‘nearly every day’ in the last two weeks, 19.2% (n = 87) reported having these thoughts ‘more than half of the days’, and 11.9% (n = 54) reported having these thoughts ‘several days’. Only 20.1% of students stated they did not have these thoughts over the time frame (n = 91).

For those who did report having suicidal thoughts in the last two weeks (n = 361), they were asked whether they had a plan to act on these thoughts. 72.3%% (n = 327) had no plans or intent to act upon these thoughts, 15.9% (n = 72) had a rough plan or some intent, and 10.4% (n = 47) had a specific plan or intent. Further analyses revealed a significant association between age and suicidal thoughts (χ (6) = 17.134, *p* = 0.009)—see Table 2.

A logistic regression was performed to identify significant predictors of suicidal thoughts. Overall, the regression model had a good fit (c^2^ (1) = 138.013, *p* < 0.001), explaining 33.4% of the variation (Nagelkerke R^2^). Depression scores (*p* < 0.001), suicide ideation (SIDAS; *p* < 0.001), and quality of life (*p* < 0.001) were significant predictors of suicidal thoughts. Table 3 provides further details.

### 3.4. Potential Risk Factors

#### 3.4.1. Suicide Literacy

The mean score for students on the LOSS was 4.46 (SD = 1.40) out of a maximum score of 9 (range = 0–12). Total scores of correct answers in the current sample ranged from 0 (0.2% of the sample) to 8 (1.3%), out of a possible 9. Most commonly, students answered four (28.1% of the sample) or five (28.3%) of the questions correctly. No significant age differences were identified. Individual responses are provided in Appendix A Table A3.

#### 3.4.2. Depressive Symptoms

The mean score for students on the PHQ-9 was 11.10 (SD = 5.91) out of a maximum score of 27 (range = 0–27). This is considered ‘moderate’ depression. A Spearman’s Rank analysis indicated a statistically significant weak, positive correlation between age and depressive scores, with depression scores increasing with age (*r*_s_ (452) = 0.101, *p* < 0.05).

#### 3.4.3. Hopelessness

The mean score for students on the Brief-H-Pos was 3.42 (SD = 1.79) out of a maximum score of 10 (range = 0–10), indicating high levels of hopelessness. Almost two-thirds (62.4%; n = 282) disagreed or strongly disagreed with the statement “*the future seems to me to be hopeful and I believe that things are changing for the better*”. Similarly, 65.0% (n = 294) disagreed or strongly disagreed with the statement “*I feel that it is possible to reach the goals I would like to stive for*”. Further details are provided in Appendix A Table A4. A one-way ANOVA indicated a significant association between age and hopelessness scores, (*F* (2, 449) = 3.514, *p =* 0.031). Post hoc analyses revealed that 15–16-year-olds were significantly more likely to be hopeful compared to 17+ year olds (0.55 ± 0.21, *p* = 0.026).

#### 3.4.4. Quality of Life

The mean score for students on the CHU9D was 19.87 (SD = 8.04) out of a maximum score of 40 (range = 0–36). Responses to individual items are presented in Appendix A Table A5.

A one-way ANOVA indicated a significant association between age and quality of life scores, (*F* (2, 449) = 5.927, *p* = 0.003). Post hoc analyses revealed that 13–14-year-olds were significantly more likely to report higher quality of life than 15–16-year-olds (2.52 ± 0.98, *p* = 0.029) and 17+ year olds (4.15 ± 1.22, *p* = 0.26).

#### 3.4.5. Help-Seeking Intentions

The mean score for students on the GHSQ was 16.27 (SD = 10.35) out of a maximum score of 35 (range = 1–33). Most commonly, students said they would seek help from a sibling (mean = 2.81; SD = 1.85) or online (mean = 2.58; SD = 2.06). They were least likely to seek help from a teacher (mean = 1.81; SD = 2.20) or counsellor (mean = 2.13; SD = 2.08).

A one-way ANOVA indicated a significant association between age and help-seeking intentions, (*F* (2, 449 = 22.59, *p* <0.001). Post hoc analyses revealed that help-seeking intentions decreased with age; 13–14-year-olds were significantly more likely to seek help than 15–16-year-olds (6.23 ± 1.22, *p* = < 0.001) and 17+-year-olds (10.04 ± 1.52, *p* = < 0.001), while 15–16-year-olds were more likely to seek help than 17+-year-olds (3.80 ± 1.23, *p* = 0.006).

## 4. Discussion

The current study aimed to explore rates of suicidal ideation and the associated risk factors among female secondary school students in Iraq. Across the current sample, students reported ‘moderate’ levels of depression and high levels of hopelessness. Students were most likely to seek help from a sibling or online and least likely to seek help from a teacher or counsellor. Quality of life was associated with age, with higher age groups (17+) reporting lower levels of quality of life, compared to 13–14-year-olds. Finally, depression, quality of life and hopelessness significantly predicted suicidal ideation among female students. Adolescent suicidal behaviour is a neglected public health issue, especially in middle- and low-income countries, and there is a greater need for more attention and research exploring suicidal ideation among students in Iraq. Furthermore, females have been shown to be disproportionately affected by suicidal ideation and/or behaviours, with societal factors and pressures placing a bigger burden on female adolescents.

Little research has been done into the mental health of Iraqis; as such, the prevalence of suicidal ideation and associated risk factors requires urgent attention and effective public health initiatives, particularly targeting youth. One national survey conducted in 2007 (Iraq Mental Health Survey; IMHS) [24], with 4332 respondents showed anxiety disorders to be most common (13.8%), followed by major depressive disorder (7.2%). Similar findings were reported in the current study, with the most common mental health diagnosis reported by students being anxiety (42%), followed by depression (22.1%) and obsessive-compulsive disorder (OCD; 16.5%). Given that the IMHS highlighted increasing lifetime prevalence of most disorders across generations, and with individuals younger than 18 years making up almost half the population, addressing the mental health of youth in Iraq is imperative [25].

Over the last two decades, past conflicts, war and economic hostilities have impacted upon the mental health of many Iraqi youth and their families. Exposure to violence, displacement and loss of family have been identified as key factors contributing to poor mental health, in particular PTSD. Research indicates that nearly one in five Iraqi children exhibits symptoms of PTSD, with rates significantly higher in areas most affected by conflict [26]. Furthermore, the ongoing instability and lack of access to mental health services exacerbates these issues, leaving many youths without support [12]. Interestingly, there was no link found between PTSD and mental health in the current study; however, this may be due to the low base rate of students with a PTSD diagnosis. Future research should, therefore, build upon this to further explore the link with PTSD-related symptoms and suicidal ideation among this population.

The current study also highlights high rates of suicidal ideation among Iraqi female secondary school students. Schools provide an important opportunity for suicide prevention and intervention, yet in the current study students disclosed they would be least likely to seek support from teachers. Most teachers in Iraq receive no training on mental health during their teaching career, and there is a high unmet need for school-based mental health interventions [27]. Students seeking help from siblings or online resources, thus, provides an important avenue to explore, highlighting the specific needs of female students and ensuring the quality and standards of that level of support online is evident to protect those young people seeking support. Despite the obvious need for school-based suicide prevention and mental health intervention, implementing these programmes in Iraq may face significant challenges, including cultural stigma and limited resources [28]. This challenge may further be amplified for female adolescents, who face additional societal pressures surrounding the role of a woman [14]. As such, the existing stigma surrounding suicidal ideation and behaviours is also important to address to better support Iraqi adolescents.

### 4.1. Limitations

While the current study provides novel insights into the rates of suicidal ideation and associated risk factors of female Iraqi youth, there are some limitations that need to be acknowledged. Firstly, this study examined students across four schools in one area in Iraq, which limits the generalisability of the findings. The sample was relatively small, and while cross-sectional studies are efficient and inexpensive, they have limitations regarding causal inference and temporal relationships, impacting their overall rigor [29]. As the study was conducted between August and December, the shift between season changes may have been associated with changes in mood and the development of Seasonal Affective Disorder (SAD) [30]. Furthermore, the validity of translated measures cannot be assumed, as well as the validity of research findings, since stigma may have been an issue for some Iraqi adolescents when answering questions regarding suicide. Additionally, regression models identified key predictors, such as depression and hopelessness, highlighting their significant roles. However, much variance remains unexplained, which may suggest the involvement of additional psychological, social or contextual factors not captured in the model. This is something future research can explore looking at capturing longitudinal data with more measures to better understand the complexity of suicidal ideation within Iraqi youth. Finally, the lack of PTSD data, despite its known prevalence in conflict-exposed youth in Iraq, limited the findings. Future research should include rigorous research designs, larger sample size and translated psychometric measures that have been validated within Iraqi schools.

#### Implications for Health and Policy

Findings from the current study highlight the need for mental health education and training, specifically suicide prevention, for school staff, supporting the necessity for and appropriateness of suicide prevention programmes for Iraqi youth. There is an urgent need for a national suicide prevention policy for Iraqi adolescents, particularly females, as well as ensuring appropriate funding allocation for mental health services to ensure access, especially in schools. The stigma around mental health and suicidal ideation, especially for females, is a major barrier to seeking help; thus, public health campaigns are needed to reduce stigma and promote open discussions about mental health. These campaigns should be sensitive to cultural norms but work towards normalising mental health care and encouraging early intervention. Finally, the ongoing conflict, economic instability and past trauma many Iraqi youths have faced has significantly impacted on mental health and suicidal ideation. Policies should, therefore, include specific mental health support for those young people affected by conflict, displacement and family loss, integrating trauma-informed care into both mental health and educational services.

## 5. Conclusions

The current study explored rates of suicidal ideation and associated risk factors among female secondary school students in Baghdad, Iraq, a previously overlooked public health concern, particularly among low and middle-income countries. Findings reveal that Iraqi secondary school students experience moderate levels of depression and high levels of hopelessness, both of which are significant predictors of suicidal ideation. The reluctance of students to seek help from teachers emphasises the importance of developing targeted school-based interventions to support young people’s mental health. Increasing research and attention in this area is vital to not only improving the mental health of students in Iraq but also reduce the stigma around mental health and suicide.

## Figures and Tables

**Table 1 healthcare-13-01260-t001:** A logistic regression examining predictors of high-risk levels of suicidal ideation.

Total Scores	β	S.E.	Wald	Significance	Exp (B)	95% CI Lower	95% CI Higher
Depression	0.095	0.028	11.825	<0.001 **	1.100	1.042	1.161
Help-seeking	0.010	0.016	0.403	0.525	1.010	0.979	1.043
Quality of life	−0.066	0.022	8.532	0.003 **	0.937	0.896	0.979
Suicide Literacy	−0.044	0.113	0.151	0.698	0.957	0.767	1.195
Hopelessness	0.166	0.082	4.061	0.044 *	1.180	1.005	1.387

* <0.05. ** <0.01.

**Table 2 healthcare-13-01260-t002:** Rates of reported suicidal thoughts by age.

Age	Not at All (n; %)	Several Days (n; %)	More than Half (n; %)	Nearly Everyday (n; %)
13–14 years (n = 85)	12 (14.1%)	4 (4.7%)	14 (16.5%)	55 (64.7%)
15–16 years (n = 307)	64 (22.6%)	35 (12.3%)	61 (21.5%)	123 (43.4%)
17+ years (n = 89)	15 (17.8%)	15 (17.8%)	12 (14.3%)	42 (49.4%)

**Table 3 healthcare-13-01260-t003:** A logistic regression examining predictors of suicidal thoughts.

Total Scores	β	S.E.	Wald	Significance	Exp (B)	95% CI Lower	95% CI Higher
Depression	0.194	0.028	47.476	<0.001 ***	1.214	1.149	1.283
Help-seeking	−0.003	0.013	0.058	0.809	0.997	0.972	1.023
Quality of life	0.068	0.019	12.342	<0.0001 ***	1.071	1.031	1.112
Suicide Literacy	−0.060	0.096	0.390	0.532	0.942	0.779	1.137
Suicide ideation (SIDAS)	−0.079	0.014	33.817	<0.001 ***	0.924	0.899	0.949
Hopelessness	−0.130	0.082	2.530	0.112	0.878	0.747	1.031

*** <0.001.

## Data Availability

The datasets generated and/or analysed during the current study will be made available at https://dese.ai/medicaldatabank/ (accessed on 21 January 2025).

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
