# Peer review of "Rates of Suicide Ideation and Associated Risk Factors Among Female Secondary School Students in Iraq"

_healthcare, 2025, doi:10.3390/healthcare13111260_

Round 1

Reviewer 1 Report

Comments and Suggestions for Authors

I have reviewed the manuscript titled “Rates of Suicide Ideation and Associated Risk Factors Among Female Secondary School Students in Iraq.” The paper addresses a critical and under-researched area in youth mental health within a politically and socially complex context. The topic is highly relevant and has strong implications for public health and education policy in Iraq. However, several methodological and conceptual issues require attention to enhance the manuscript’s rigour, clarity, and clinical relevance. The cross-sectional design limits causal interpretation. This should be clearly stated in both the abstract and discussion. All data were self-reported using standardized scales, yet without clinical interviews. This raises concerns about diagnostic accuracy, especially in a context where stigma may lead to underreporting. While translation procedures were detailed (p. 5), the lack of psychometric validation of the Arabic versions in this population is a significant gap. Sampling from only four schools in Baghdad may not reflect broader regional or socioeconomic diversity. More information on school selection is needed. Seasonal timing of data collection (August–December) could affect mood-related variables, yet this is not discussed. The survey could have benefited from qualitative input to capture cultural and contextual nuances influencing suicidal ideation. The finding that students were least likely to seek help from teachers or counselors (p. 8) needs deeper exploration. Cultural and systemic factors behind this pattern should be discussed more fully. Regression models identified key predictors like depression and hopelessness, but much variance remains unexplained. Other contextual variables (e.g., trauma history) should be considered in future work. The limitation section should explicitly mention the lack of PTSD data, despite its known prevalence in conflict-exposed youth in Iraq. The conclusion would be stronger with actionable recommendations, such as adapting international school-based prevention models to the Iraqi context.

Comments on the Quality of English Language

 The English could be improved to more clearly express the research.

Author Response

We thank the reviewers for taking their time to read our paper and provide us the opportunity to make the changes and suggestions below. We have made changes within the manuscript and highlighted these in yellow throughout.

Review 1

  • I have reviewed the manuscript titled “Rates of Suicide Ideation and Associated Risk Factors Among Female Secondary School Students in Iraq.” The paper addresses a critical and under-researched area in youth mental health within a politically and socially complex context. The topic is highly relevant and has strong implications for public health and education policy in Iraq. However, several methodological and conceptual issues require attention to enhance the manuscript’s rigour, clarity, and clinical relevance.

Thank you for the positive comments and highlighting some of the issues that need addressing.

  • The cross-sectional design limits causal interpretation. This should be clearly stated in both the abstract and discussion.

This point has been added to both the abstract and discussion.

  • All data were self-reported using standardized scales, yet without clinical interviews. This raises concerns about diagnostic accuracy, especially in a context where stigma may lead to underreporting. While translation procedures were detailed (p. 5), the lack of psychometric validation of the Arabic versions in this population is a significant gap.

We agree with the reviewers and have alluded to this in the limitations and future research is needed to address this.

  • Sampling from only four schools in Baghdad may not reflect broader regional or socioeconomic diversity.

We agree with the reviewers and have expanded on this within the limitations: “The sample was relatively small, and while cross-sectional studies are efficient and inexpensive, they have limitations regarding causal inference and temporal relationships, impacting their overall rigor.30”

  • More information on school selection is needed. Seasonal timing of data collection (August–December) could affect mood-related variables, yet this is not discussed.

Within the design we have described the school selection. Four schools were selected across two regions of Baghdad to ensure representation of schools across the city. Seasonal timing was not discussed within the study and we have added this within the limitations of the study: “As the study was conducted between August and December, shift between season changes may have been associated with changes in mood and the development of Seasonal Affective Disorder (SAD).31”

  • The survey could have benefited from qualitative input to capture cultural and contextual nuances influencing suicidal ideation.

We agree with the reviewers comments and this work is planned for future research.

  • The finding that students were least likely to seek help from teachers or counsellors (p. 8) needs deeper exploration. Cultural and systemic factors behind this pattern should be discussed more fully.

We have added more information within the discussion: “

  • Regression models identified key predictors like depression and hopelessness, but much variance remains unexplained.

We have now added more information into the discussion:

“Additionally, regression models identified key predictors such as depression and hopelessness, highlighting their significant roles. However, much variance remains unexplained which may suggest the involvement of additional psychological, social or contextual factors not captured in the model. This is something future research can explore looking at capturing longitudinal data with more measures to better understand the complexity of suicidal ideation within Iraqi youth.”

  • Other contextual variables (e.g., trauma history) should be considered in future work.

We agree with the reviewers comments and a study is currently underway including psychometric measures around PTSD and trauma.

  • The limitation section should explicitly mention the lack of PTSD data, despite its known prevalence in conflict-exposed youth in Iraq.

The following text has been added to the limitations section: “Finally, the lack of PTSD data, despite its known prevalence in conflict-exposed youth in Iraq, limited the findings. Future research should include rigorous research designs, larger sample size and translated psychometric measures that have been validated within Iraqi schools.”

  • The conclusion would be stronger with actionable recommendations, such as adapting international school-based prevention models to the Iraqi context.

The following text has been added to the conclusion: “Future work should include adapting international school-based prevention models to the Iraqi context.”

Reviewer 2 Report

Comments and Suggestions for Authors

See enclosed comments.

Comments on the Quality of English Language

See enclosed comments.

Author Response

Review 2

General Comments to Authors: The field of suicide prevention is very expansive, also including purely clinical approaches and societal structural approaches. This study on psychiatric factors fills in the triangle. The mental health conditions assessed and clinical instruments used might appeal to the readership. Some light revision needs to occur up front, e.g., explicit mention of policy conclusions, role of social isolation with COVID. The terse mention of statistical tests in the Methods section could benefit from more description/unpacking. Other study-related fill in needs to occur in the Methods. Grammatical corrections are indicated below.

We thank the reviewer for the positive and useful comments which we have addressed below.

Specific Comments to Authors:

  • 1, Para. 1, line 38 (Abstract): Add a new sentence or rewrite an existing one in the Abstract to explicitly indicate the “health policy” changes being suggested at the end of your paper.

The following text has been added to the conclusion section of the abstract: Future policies should include specific mental health support for those young people affected by conflict, displacement and family loss, integrating trauma-informed care into both mental health and educational services.”

  • 2, Para. 4, line 72: Mention social isolation connected with COVID-19 and add an appropriate reference.

We have added further information and references as requested: “Additionally, the Covid-19 pandemic exacerbated the global mental health crisis by increasing social isolation, linked to higher risks of depression, anxiety, and cognitive decline.10,11”

  • 3, Para. 4: line 114: Indicate whether all the surveys, both primary instruments and more specialized questionnaires, were conducted on paper.

We have now added information into the method section to clarify this point:

“Student were invited to complete paper-based surveys in school, which included all measures stated below.”

  • line 121: Add in a numeric figure+denominator/percentage for the students leaving before starting and not completing the questionnaires.

The percentage of student who chose to leave has been added. We have now clarified this within the text: “Some students chose to leave before starting or did not complete the questionnaire (n=27; 6%).”

  • 4, Para. 5, line 174: “Suicide literacy” is a specialized term that requires a sentence of explanation. What does “literacy” mean in this context?

More information has been added about suicide literacy: “Suicide literacy was assessed using an adapted version of the Literacy of Suicide Scale (LOSS) – short form to ascertain participants knowledge about suicide.21”

  • 4, Para. 6, line 179: Was the pilot study conducted in one school or several schools? Approximately how many students or classes/ave. class size were in the pilot?

The pilot survey was shared with seven school aged young people known to the team of researchers, who were from a similar age group to the potential participants completing the survey, to check if the translation was fit for purpose.  We have added the number of young people to the text. “The young people reviewed if the questions would be understood following the translation of the survey.  A pre-testing pilot was conducted to test the translation of the survey from English to Arabic on a small group similar to the target population (n=7).”

  • 5, Para. 1, lines 187-8: This string of statistical test mentions needs unpacking. Please explain the type of exploration each test is being used for. Can explain in terms of categories of explorations/investigations or sets of tests.

Thank you to the reviewer for their comment, we have now revised the relevant section on page 5 to add further clarity:

“This was an exploratory analysis; whereby descriptive statistics were conducted to identify the socio-demographics of the sample and the factors characteristic of students experiencing suicide ideation. To achieve this, we conducted the following set of analyses. Descriptive analysis (e.g., means, standard deviations, frequencies, percentages) were used to characterize the overall sample in terms of age, gender, ethnicity, academic status, and other socio-demographic variables. Next, chi-squared analyses were used to assess associations between categorical variables (e.g., gender and suicidal thoughts), as well as independent samples t-tests and one-way ANOVAs to compare means of continuous variables (e.g., levels of stress, social support) across groups defined by the presence or absence of suicide ideation or across multiple categorical groups (e.g., academic year). Association analyses using Spearman’s Rank correlation were also employed to examine relationships between continuous or ordinal variables (e.g., perceived stress and suicidal ideation scores). Regression analyses were then conducted with suicide ideation and suicidal thoughts as outcome variables to identify significant predictors. We did not conduct Bonferroni corrections for multiple comparisons based on recommendations from Armstrong22 and Rothman23 that corrections for multiple comparisons in exploratory studies are not required, due to the increased likelihood of Type 2 errors”

  • 6, Para. 1, line 215: Please double-check whether the mental health diagnoses being referred to are previous or current.

This has been checked and we are alluding to …. mental health diagnosis.

  • 6, Para. 2, line 224: Add a comment with reference on whether this high level of suicidal ideation is seen in other countries. (Can either place here or in Discussion)

We have added this information in the discussion:

  • 9, End Items: Add: (1) Funding; (2) Institutional Review Board statements

This information has been added.

Grammatical: A third party versed in English grammar is suggested for the following categories of corrections:

All of the grammatical corrections have been completed and highlighted within the text.

22.(1) Inappropriate commas, e.g., line 19: among Iraqis, is rising -> among Iraqis is rising

22.(2) Gaps, e.g., line 20: to trauma, economic hopelessness -> to trauma, economic hopelessness

22.(3) Inflections, e.g., line 114: Student were invite to complete -> Students were invited to complete; line 121: questionnaire. -> questionnaires

22.(4) Unnecessary hyphens, e.g., line 292: adolescents.- -> adolescents.

22.(5) Unnecessary underlines, e.g., line 314: further -> further

Reviewer 3 Report

Comments and Suggestions for Authors

You have conducted a study on suicidal ideation. I think this study is important in terms of adolescent mental health, and the results of the study can be used as important evidence for suicide prevention. This paper is well organized overall and I read it with great interest. I think the introduction has a good logical flow. I think the topic of the paper is within the scope of publication in this journal, but I suggest a few minor revisions as follows:

. It would be better to show 4.1. and 4.2. before limitations and Implications for Health and Policy in Discussion for readability.

Author Response

You have conducted a study on suicidal ideation. I think this study is important in terms of adolescent mental health, and the results of the study can be used as important evidence for suicide prevention. This paper is well organized overall and I read it with great interest. I think the introduction has a good logical flow. I think the topic of the paper is within the scope of publication in this journal, but I suggest a few minor revisions as follows:

  • It would be better to show 4.1. and 4.2. before limitations and Implications for Health and Policy in Discussion for readability.

We thank the review for positive feedback on the papers. We think the paper follows the format they suggest. However, if this is not the case please let us know.

Round 2

Reviewer 1 Report

Comments and Suggestions for Authors

The authors have completely addressed all my comments, and I have no further concerns. Therefore, I recommend accepting the paper.

Comments on the Quality of English Language

 The English could be improved to more clearly express the research.